# Cryo-electron tomographic investigation of native hippocampal glutamatergic synapses

Aya Matsui[1,2†], Cathy Spangler[2†], Johannes Elferich[3], Momoko Shiozaki[4], Nikki Jean[4], Xiaowei Zhao[4], Maozhen Qin[2], Haining Zhong[2], Zhiheng Yu[4], Eric Gouaux[1,2]*

[1]Howard Hughes Medical Institute, Oregon Health and Science University, Portland, United States; [2]Vollum Institute, Oregon Health and Science University, Portland, United States; [3]RNA Therapeutics Institute, University of Massachusetts Chan Medical School, Howard Hughes Medical Institute, Worcester, United States; [4]Howard Hughes Medical Institute, Janelia Research Institute, Ashburn, United States

## eLife assessment

This **fundamental** study demonstrates a novel method for imaging glutamate receptors in situ via cryo-ET. The use of cutting-edge methods is well-described and is **compelling**. This paper is broadly relevant to biophysicists and neuroscientists.

*For correspondence:
gouauxe@ohsu.edu

†These authors contributed equally to this work

Competing interest: The authors declare that no competing interests exist.

**Abstract** Chemical synapses are the major sites of communication between neurons in the nervous system and mediate either excitatory or inhibitory signaling. At excitatory synapses, glutamate is the primary neurotransmitter and upon release from presynaptic vesicles, is detected by postsynaptic glutamate receptors, which include ionotropic AMPA and NMDA receptors. Here, we have developed methods to identify glutamatergic synapses in brain tissue slices, label AMPA receptors with small gold nanoparticles (AuNPs), and prepare lamella for cryo-electron tomography studies. The targeted imaging of glutamatergic synapses in the lamella is facilitated by fluorescent pre- and postsynaptic signatures, and the subsequent tomograms allow for the identification of key features of chemical synapses, including synaptic vesicles, the synaptic cleft, and AuNP-labeled AMPA receptors. These methods pave the way for imaging brain regions at high resolution, using unstained, unfixed samples preserved under near-native conditions.

## Introduction

Chemical synapses underpin the majority of communication sites between neurons and are comprised of a presynaptic terminal, a synaptic cleft, and a postsynaptic apparatus (*Südhof and Malenka, 2008*). Signal transduction at synapses involves voltage-dependent and calcium-stimulated release of neurotransmitters from presynaptic vesicles, typically raising the concentration of neurotransmitters in the cleft as much as $10^5$-fold (*Clements et al., 1992*). The bolus of neurotransmitter in turn promotes activation of synaptic and perisynaptic receptors, which are primarily located on postsynaptic specializations, thereby inducing depolarization, calcium entry into dendrites, and propagation of the neuronal impulse. At glutamatergic synapses, the neurotransmitter signal is quenched by clearance from synaptic and extracellular spaces by binding to and subsequent uptake via sodium-coupled transporters. Despite the central roles that chemical synapses play in the development and function of the nervous system, and their association with neurological disease and dysfunction, the molecular

structure of the synapse has remained elusive, largely due to challenges associated with imaging at high resolution.

The first images of chemical synapses were derived from electron microscopy studies of fixed and stained neuronal and neuromuscular synapses (*Palade and Palay, 1954*). Chemical fixation, dehydration, and heavy metal staining during the sample preparation process, in combination with early electron microscopes and recording methods, severely constrained the resolution of the images and likely distorted molecular features. Nevertheless, it was possible to make out presynaptic vesicles, the synaptic cleft, and the postsynaptic density. Subsequent electron microscopy studies, using both cultured neurons and brain tissue-derived samples, together with imaging studies of biochemically isolated synaptosomes, have provided progressively higher-resolution visualizations of synapses and their surrounding structures (*Heuser and Reese, 1973*; *Harris and Stevens, 1989*; *Schikorski and Stevens, 1997*). More recently, electron microscopy experiments, together with light microscopy studies, suggest that in excitatory, glutamatergic synapses, there is a transsynaptic organization of presynaptic and postsynaptic components that aligns sites of transmitter release and detection (*Chen et al., 2008*; *Tang et al., 2016*; *Tao et al., 2018*; *Martinez-Sanchez et al., 2021*; *Cole and Reese, 2023*).

Here, we report approaches for imaging glutamatergic synapses from the CA1 region of the hippocampus, a well-studied region of the brain that plays crucial roles in memory and learning and that includes the pyramidal cell, CA3-CA1 Schaffer collateral synapses. To enable the identification and visualization of glutamatergic synapses, we developed several key methods. First, we generated an engineered mouse line to pinpoint excitatory synapses via fluorescence methods. Second, we developed a GluA2-specific antibody fragment (*Giannone et al., 2010*) AuNP conjugate (*Azubel and Kornberg, 2016*) to localize AMPA subtype glutamate receptors. Third, we optimized sample preparation from unfixed hippocampal brain tissue, together with the application of cryo-protectants and high-pressure freezing (HPF). With the frozen slices in hand, we next employed cryo-focused ion beam, scanning electron microscopy (cryo-FIB/SEM) milling methods (*Marko et al., 2007*; *Rigort et al., 2012*; *Kelley et al., 2022*) to produce thin lamella, in turn using pre- and postsynaptic fluorescence signals to guide imaging by cryo-electron tomography (cryo-ET). By combining all of these approaches, we created an experimental workflow that allows us to image excitatory synapses from unfixed, chemically unstained brain tissue by cryo-ET at near nanometer resolution, thereby opening the door to the direct localization of molecules involved in synapse structure, organization, and function.

## Results

### Anti-GluA2 15F1 Fab conjugated to a single gold nanoparticle

To provide an electron-dense marker for labeling individual, GluA2-containing AMPARs at glutamatergic synapses in cryo-electron tomograms, we developed a strategy to label the anti-GluA2, 15F1 Fab (*Giannone et al., 2010*; *Rigort et al., 2012*) with a single AuNP based on previously determined methods to label antibody fragments with AuNPs (*Azubel et al., 2019*; *Elferich et al., 2021*). To accomplish this, we synthesized uniform ~3 nm 3-mercaptobenzoic acid (3-MBA) thiolate-protected gold nanoparticles by established methods (*Azubel and Kornberg, 2016*). AuNPs were coupled to an anti-GluA2 15F1 Fab engineered with an extended heavy chain containing a single free cysteine, thus enabling a thiol exchange reaction with 3-MBA and covalent coupling of the Fab to the AuNP (*Figure 1A*, *Figure 1—figure supplement 1*). By optimizing the relative ratio of Fab and AuNP, we maximized the formation of a 1:1 Fab:AuNP species (*Figure 1B*). Following Fab conjugation, AuNPs were PEGylated to minimize aggregation of the nanoparticles (*Figure 1—figure supplement 1*).

The anti-GluA2 15F1 Fab-AuNP conjugate remains fully capable of GluA2 binding, shifting a tetrameric GluA2 peak to an extent consistent with 1:1 Fab:GluA2 binding by fluorescence-detection size-exclusion chromatography (FSEC) (*Kawate and Gouaux, 2006*; *Figure 1C*). Taking advantage of an affinity tag included at the C-terminus of the Fab, the anti-GluA2 Fab-AuNP conjugate was used to affinity purify native GluA2-containing AMPARs from mouse hippocampal tissue (*Zhao et al., 2019*; *Yu et al., 2021*). The isolated receptors were then visualized by single particle cryo-EM to investigate the stoichiometry of the 15F1 Fab-AuNPs bound to native AMPARs and to also determine the spatial positioning of the AuNPs relative to the receptor. Previous work from our group employing a similar anti-GluA2 15F1 Fab-directed purification of native hippocampal

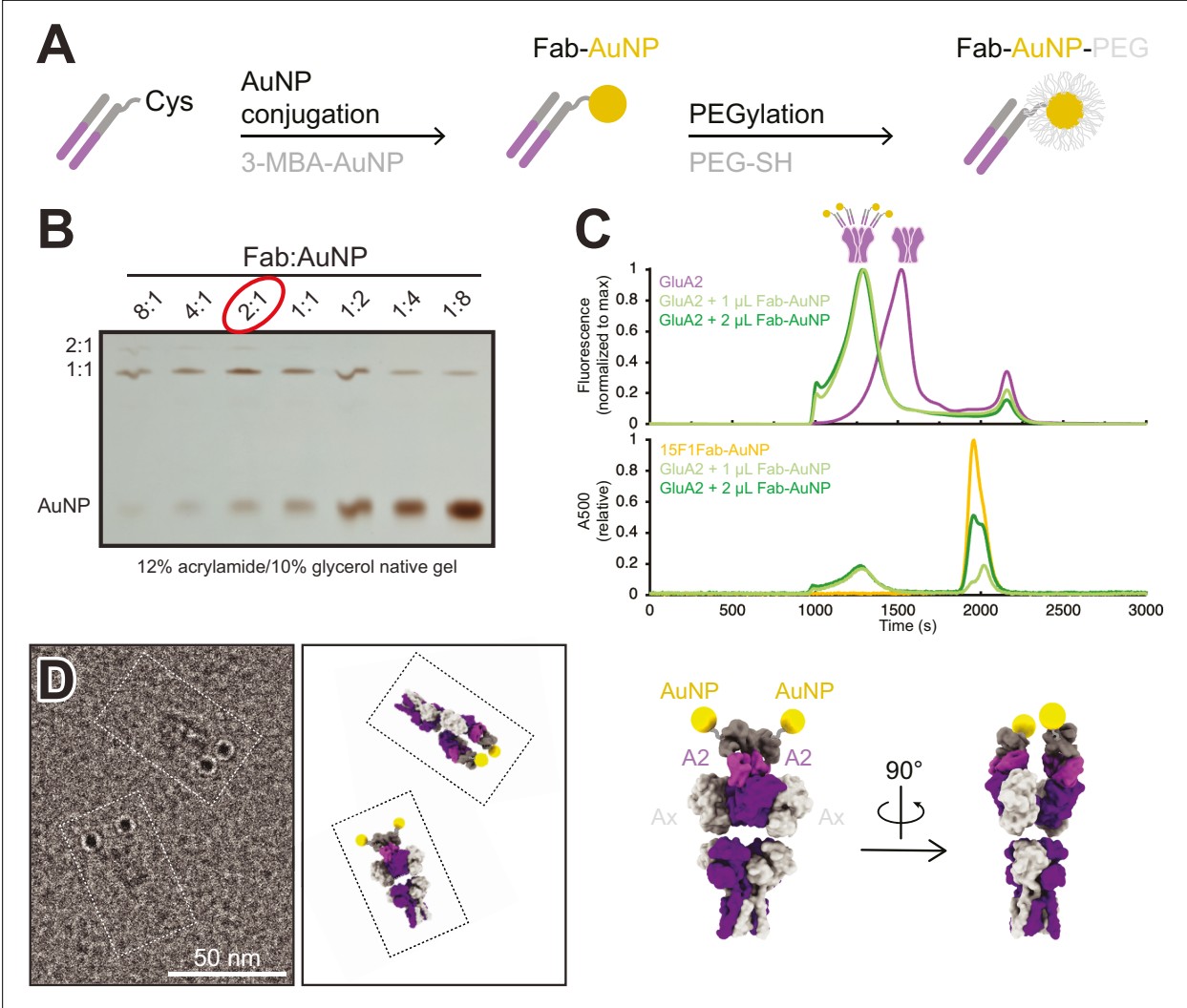

**Figure 1.** Development and characterization of anti-GluA2 Fab conjugated to gold nanoparticle (AuNP). (**A**) Conjugation strategy for covalent linkage of AuNP to anti-GluA2 Fab. (**B**) Test-scale conjugations of Fab and AuNP at various Fab:AuNP ratios, run on native 12% acrylamide/10% glycerol gel. (**C**) Normalized fluorescence-detection size-exclusion chromatography (FSEC) traces of GFP-tagged GluA2 mixed with 1–2 μL of Fab-AuNP conjugate, measured at an excitation/emission of 480/510 nm (top; corresponding to GFP-tagged GluA2) or an absorbance at 500 nm (bottom; corresponding to AuNP). (**D**) Snapshot from single particle cryo-electron microscopy micrograph image showing two individual Fab-AuNP bound native mouse hippocampus AMPARs next to models depicting possible orientational views seen in image (PDB: 7LDD) (left). Model of anti-GluA2 Fab-AuNP bound to AMPAR with GluA2 in the B and D positions (right).

The online version of this article includes the following source data and figure supplement(s) for figure 1:

**Source data 1.** PDF file containing original gels for *Figure 1B*, indicating relevant bands.

**Source data 2.** Original files for gel analysis are displayed in *Figure 1B*.

**Figure supplement 1.** Preparation of PEGylated AuNP-15F1 Fab conjugate.

**Figure supplement 1—source data 1.** PDF file containing original gels for *Figure 1—figure supplement 1C*, indicating the relevant bands.

**Figure supplement 1—source data 2.** Original files for gel analysis are displayed in *Figure 1—figure supplement 1C*.

**Figure supplement 2.** Single particle cryo-electron microscopy (cryo-EM) of Fab-gold nanoparticle (AuNP) bound to native mouse hippocampal AMPAR.

AMPARs revealed that the majority of GluA2-containing AMPARs in the mouse hippocampus contain two GluA2 subunits, occupying the B and D positions of the AMPAR complex (*Yu et al., 2021*). In agreement with the studies of Yu and colleagues, imaging of the 15F1 Fab-AuNP-purified AMPARs revealed that the majority of particles harbor two clear AuNP densities present at positions consistent with the 15F1 Fab binding epitope on the amino-terminal domain (ATD) of the GluA2 subunits

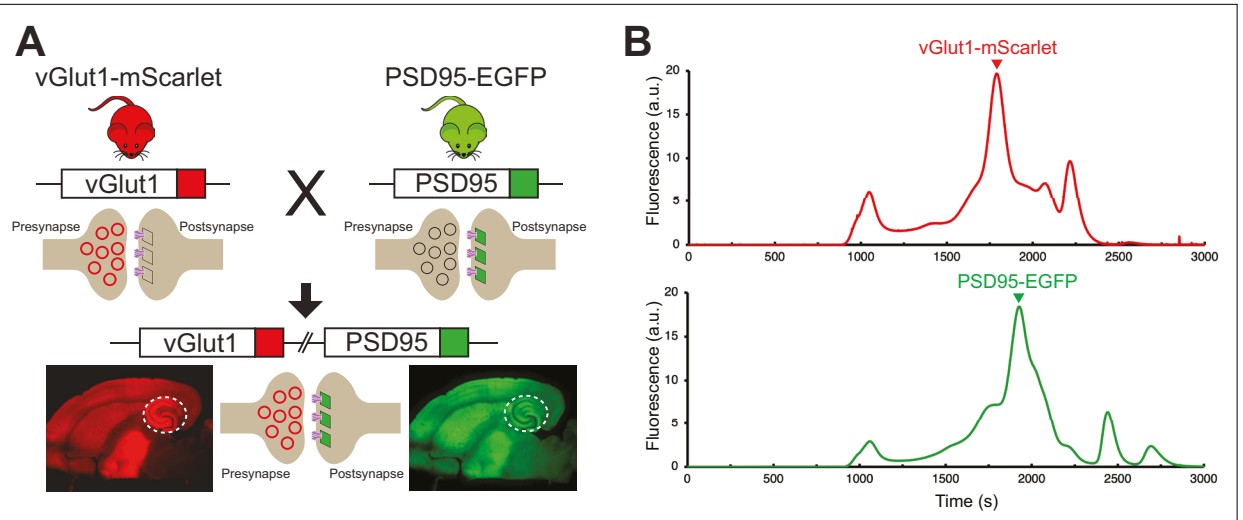

**Figure 2.** Development and characterization of transgenic mouse line for synaptic targeting. (**A**) Red fluorescent protein, mScarlet, was inserted at the C-terminus of vGlut1, which is present within synaptic vesicles at excitatory presynaptic terminals. In another mouse line, the green fluorescent protein, EGFP, was inserted at the C-terminus of PSD95, which is highly expressed at excitatory postsynaptic sites. Homozygous vGlut1-mScarlet and PSD95-EGFP mice were crossed to generate a mouse line expressing both vGlut1-mScarlet and PSD95-EGFP to facilitate synapse identification. The white circle represents the hippocampus region. (**B**) Assessment of vGlut1-mScarlet and PSD95-EGFP in solubilized mouse whole brain tissue by fluorescence-detection size-exclusion chromatography (FSEC) confirms expression of each fluorescent fusion protein.

(*Figure 1D*, *Figure 1—figure supplement 2*). We do note that a minority of particles, less than about 20%, bear only a single AuNP, which may be due to either incomplete labeling, to labeling with a Fab devoid of an AuNP, to receptors with only a single GluA2 subunit, or to the unbinding of the Fab-AuNP and perhaps, separately, the loss of the AuNP itself. Nevertheless, we speculate that the majority of singly labeled receptors are simply those with one GluA2 subunit for several reasons. Not only was the Fab-AuNP conjugate purified by gel electrophoresis, but we also employed an excess of the Fab-AuNP conjugate, and the Fab-AuNP receptor complex was additionally purified by size exclusion chromatography.

## Fluorescence-guided localization of glutamatergic synapses

To enable the reliable, targeted imaging of AMPARs at excitatory glutamatergic synapses in the context of the relatively large area of the lamella, a transgenic mouse line expressing mScarlet fused to the presynaptic vesicular glutamate transporter 1 (vGlut1) and EGFP fused to the postsynaptic density protein 95 (PSD95) was generated (*Figure 2A*). The vGlut1-mScarlet mouse line was designed in an identical manner to a previously established vGlut1-mVenus knock-in line in which the C-terminus of vGlut1 is fused to mVenus (*Herzog et al., 2011*). vGlut1 is abundantly expressed in glutamatergic synaptic vesicles, allowing for clear fluorescent visualization of excitatory presynaptic sites within tissue (*Herzog et al., 2011*). At postsynaptic sites, PSD95 is the m ost abundant scaffolding protein within the postsynaptic density (*Fortin et al., 2014*; *Cheng et al., 2006*) and thus we generated a mouse line in which PSD95 is tagged at the C-terminus with EGFP. To enable simultaneous visualization of pre-and postsynaptic sites by fluorescent imaging, the homozygous vGlut1-mScarlet and PSD95-EGFP mouse lines were bred to make a double transgenic mouse line expressing both vGlut1-mScarlet and PSD95-EGFP. Assessment of vGlut1-mScarlet and PSD95-EGFP expression by FSEC confirmed the presence of a major peak corresponding to each fusion protein in solubilized mouse whole brain tissue (*Figure 2B*). Therefore, we expect stable expression and minimal cleavage of the fusion proteins within the vGlut1-mScarlet/PSD95-EGFP mouse line. We note that the vGlut1-mScarlet/PSD95-EGFP mouse line has no apparent phenotype, and all animals exhibit wild-type growth, breeding, and overall behavior, thus leading us to conclude that their glutamatergic synapses are representative of those in wild-type animals.

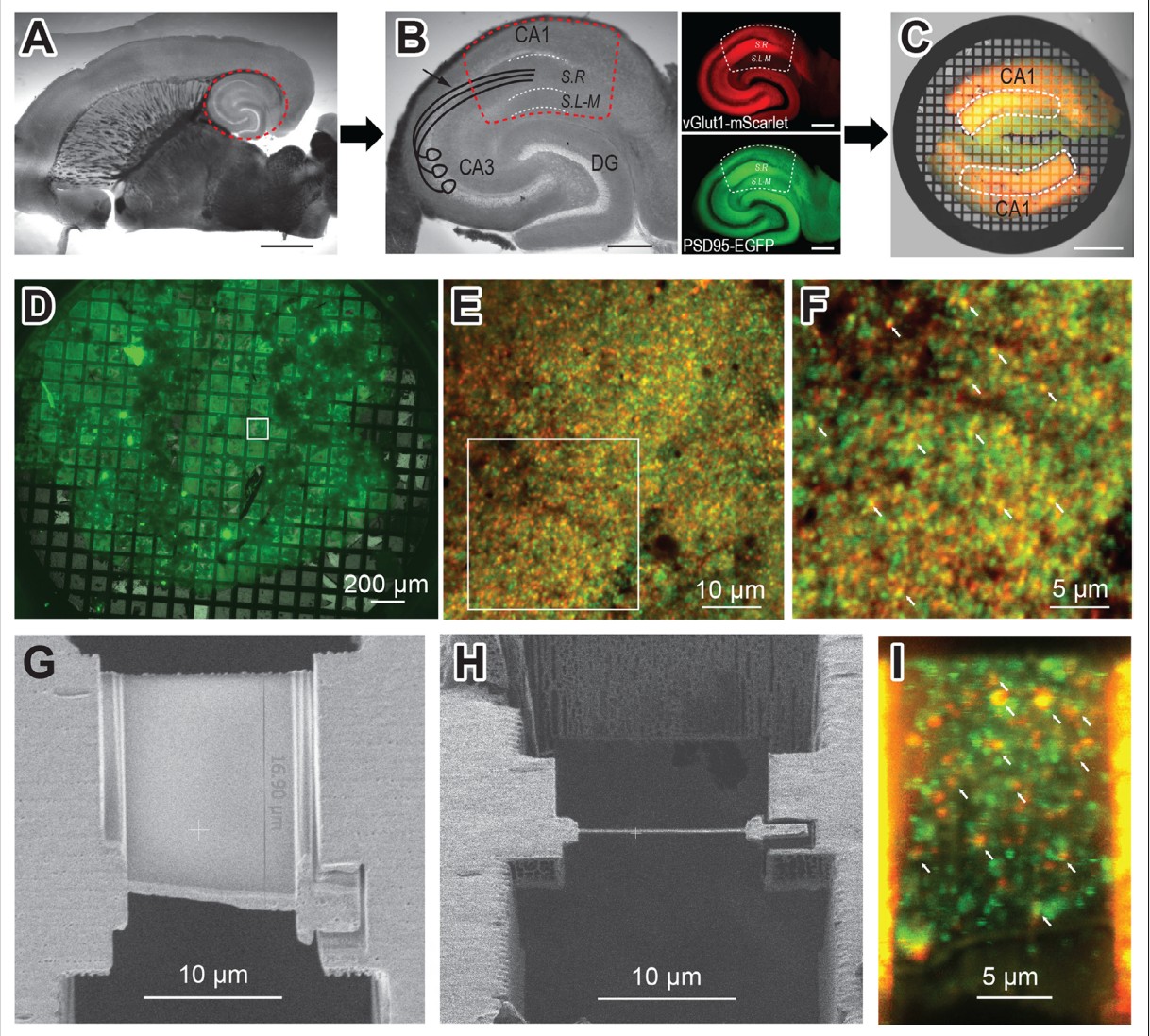

**Figure 3.** Hippocampus brain slice preparation and focused ion beam (FIB) milling of lamella. (**A**) Ultra-thin brain slice made from a vGlut1-mScarlet and PSD95-EGFP mouse. The red dashed circle indicates the hippocampus. Scale bar: 2 mm (**B**) A wide-field image of excised hippocampus brain slice. The image also illustrates CA3 axons projecting to CA1 apical dendrites in the Schaffer collateral (arrow). Fluorescent images show vGlut1-mScarlet (top) and PSD95-EGFP (bottom) from the same hippocampus slice on the left. The dashed line (red or white) area indicates the region of interest for FIB milling (CA1 region of hippocampus), which is further trimmed with a scalpel knife. S.R: Stratum Radiatum, and S.L-M: Stratum lacunosum-moleculare. Scale bar: 500 μm (**C**) Overlay image of two trimmed hippocampus brain slices on electron microscopy (EM) grid, before high-pressure freezing. The white dashed line indicates the stratum radiatum, the area with the brightest fluorescence in the CA1 region and where the Schaffer collateral projects. Scale bar: 500 μm (**D**) A wide field green fluorescence image of a sample grid after high-pressure freezing. (**E**) Cryo-confocal image from a sample grid showing both vGlut1-mScarlet (red) and PSD95-EGFP (green) puncta. (**F**) Zoomed image of area indicated in E. White arrows point to colocalization of red and green fluorescent signals and indicate potential synapses. (**G**) Scanning electron microscopy (SEM) image of the polished lamella. (**H**) FIB image of same lamella in G. (**I**) Cryo-confocal image of lamella. White arrows at fluorescence colocalization points indicate examples of regions targeted in tomography data collection.

The online version of this article includes the following figure supplement(s) for figure 3:

**Figure supplement 1.** Pre-soaking tissue with 15F1 Fab-gold nanoparticle (AuNP) blocks the majority of available GluA2 binding sites.

**Figure supplement 2.** Cryo-correlative light and electron microscopy (Cryo-CLEM) guides synaptic targeting on lamella.

## Mouse hippocampus brain slice preparation and freezing

To visualize AMPARs at glutamatergic synapses in their native context, we developed a strategy to prepare and image mouse hippocampal brain tissue by cryo-ET. Ultra-thin horizontal brain slices from the double transgenic mouse line (vGlut1-mScarlet/PSD95-EGFP) were prepared using a vibratome

set to 40 μm thickness (*Figure 3A*). Brain slices of this size are thin enough to minimize ice formation and make cryo-focused ion beam (cryo-FIB) milling less challenging, while thick enough to handle during immunolabeling and high-pressure freezing. From each horizontal brain slice, the hippocampal region was excised carefully with a scalpel knife (*Figure 3B*). These hippocampal slices were incubated with the anti-GluA2 15F1 Fab-AuNP conjugate for 1 hr at room temperature on an orbital shaker. Following the washout of the antibody-containing buffer, hippocampal slices were trimmed to isolate the CA1 region, which contains the Schaffer collateral pathway from CA3 synapses onto CA1 apical dendrites in the stratum radiatum (dashed line in *Figure 3B*; *Andersen et al., 2007*). The slices were then briefly incubated in a buffer supplemented with 20% dextran, as a cryoprotectant. Two hippocampal CA1 slices were placed on the grid bar side of an electron microscopy (EM) grid coated with a 20–30 nm thick carbon film, with apical dendrites from CA1 neurons facing toward the center of the grid (*Figure 3C*). We note that the stratum radiatum shows brighter fluorescence both in red and green channels compared to the striatum lacunosum-moleculare (*Figure 3B* insets), which helped guide targeting.

We utilized the 'waffle' method for high-pressure freezing (*Kelley et al., 2022*). At the preparation stage of the high-pressure freezer, a sample grid was placed between the flat sides of two polished and pretreated planchettes. Immediately before high-pressure freezing, gold fiducials prepared in media containing dextran and sucrose cryoprotectants were applied to the sample on the planchettes. During high-pressure freezing, the tissue was pressed in between the grid bars and frozen. Because the sample thickness is approximately equivalent to the height of the grid bars, the portion of the sample that is between the grid bars is minimally compressed during freezing. High-pressure freezing using the methods described above limited the formation of crystalline ice and results in cryo-fixed tissue without the introduction of substantial visible freezing artifacts.

## Saturation of binding sites by the 15F1 Fab-AuNP conjugate

To examine the extent to which the anti-GluA2 Fab-AuNP conjugate saturates available binding sites within the brain tissue slices, hippocampus slices were soaked with or without the 15F1 Fab-AuNP conjugate at a concentration of 40 nM for 30 min. After washout, the tissue was subsequently soaked with the 15F1 Fab labeled with Janelia Fluor 646 (40 nM) for another 30 min. We reasoned that if the 15F1 Fab-AuNP conjugate saturated all available binding sites on GluA2-containing AMPARs, further labeling of GluA2-containing AMPARs with the 15F1 Fab-Janelia Fluor 646 conjugate should be low. In tissue samples without 15F1 Fab-AuNP pretreatment, we observe clear 15F1 Fab-Janelia Fluor 646 signal that slightly overlaps with the PSD95-EGFP and vGlut1-mScarlet fluorescent signals (*Figure 3—figure supplement 1*). However, in slices previously soaked with 15F1 Fab-AuNP, the 15F1 Fab-Janelia Fluor 646 signal was substantially reduced. Based on this experiment and the established propensity of the 15F1 antibody and its fragments to label GluA2-containing AMPARs (*Giannone et al., 2010*), we expect that the 'staining' with the 15F1 Fab-AuNP used in our tissue preparation pipeline labels the majority of accessible GluA2-containing AMPARs within the tissue.

## Cryo-FIB milling of mouse hippocampus tissue

High-pressure frozen mouse hippocampus tissue slices on EM grids were loaded into an Aquilos 2 cryo-FIB/SEM microscope and milled using the 'waffle' method (*Kelley et al., 2022*). Scanning electron microscopy (SEM) images were aligned with fluorescent light microscope images of the entire sample grid and regions with maximal green and red fluorescence were identified for targeting. In the apical dendrites of CA1 neurons, there are abundant excitatory synapses, especially from CA3 axons to CA1 dendrites. Before freezing, we intentionally oriented the apical dendrites of CA1 neurons toward the center of the grid. By carrying out the FIB milling near the center of the grid, we increase the likelihood of targeting excitatory synapses instead of soma or distal dendrites (*Figure 3C–F*).

Waffle milling was most successful on grids where the grid bars were clearly visible in the FIB/SEM images, indicating sufficiently thin ice, and on grid regions where the surface was relatively smooth, as judged by FIB/SEM imaging. For each lamella, initial trench cuts were made to mill through the entire depth of the sample at the front and back edges of a lamella target site. After making trench cuts for each lamella target, typically amounting to about 10 per grid, the grids were tilted to a 20° milling angle and milled with progressively lower FIB current to a final thickness of approximately 150–250 nm (*Figure 3G and H*).

## Identification of synapses within lamellae by cryo-confocal imaging

To confirm the presence of intact synapses within lamellae and to guide synaptic targeting during cryo-ET data collection, lamellae were imaged by cryo-fluorescence confocal microscopy. Grids with milled lamellae were loaded onto a confocal microscope equipped with a cryo-correlative microscopy stage, thus maintaining the sample at cryogenic temperatures during imaging. Clear signals for both presynaptic vGlut1-mScarlet and postsynaptic PSD95-EGFP were visible on the majority of lamellae. Ideal synaptic targets on lamellae are indicated by colocalization of red presynaptic vGlut1-mScarlet and green postsynaptic PSD95-EGFP puncta, visualized as yellow overlapping spots (*Figure 3E and F*). At each lamella site, confocal fluorescence images were taken from the lamella area, typically about 20×13 µm in size. Because the lamellae were milled at a 20 degree angle, approximately eight 1 µm increment Z-stack images were taken and compiled to visualize an entire single lamella (*Figure 3I*). On an average single lamella, we typically detect at least a dozen spots with colocalized fluorescent puncta, highlighting ideal regions for synaptic targeting during subsequent cryo-ET data collection (*Figure 3I*, *Figure 3—figure supplement 2*).

## Cryo-electron tomography of hippocampal synapses

Cryo-FIB milled lamellae were imaged using a 300 keV Krios cryo-transmission electron microscope. Initially, medium magnification montage maps of each lamella were taken to guide the picking of synaptic targets. Often, clear cellular compartments containing synaptic vesicles can be identified by eye from the medium magnification EM maps. Additionally, cryo-correlative light and electron microscopy (cryo-CLEM) further guided the selection of targets (*Figure 3—figure supplement 2*). Registration of the fluorescence and electron microscopy maps was performed by alignment of manually identified lamella structural features, often based on the edge shape of the lamella visible in both the confocal reflection and EM images. Once aligned, the presynaptic vGlut1-mScarlet signal often overlaps with cellular compartments containing synaptic vesicles in the EM map, substantiating alignment of the light and EM images. Synaptic targets were picked both by manually scanning the EM map for presynaptic vesicle-looking features as well as the overlaying fluorescence signal. Typically, we selected synaptic targets oriented such that the pre-and postsynaptic terminals are side by side with a visible synaptic cleft in between (*Figure 4A and B*). We picked as many as 10–30 tomography targets within a single 20 µm × 13 µm lamella depending upon the presence of synaptic features and the quality of lamella ice. The tilt series of the targets were acquired using a grouped dose-symmetric scheme (*Hagen et al., 2017*) starting normal to the lamella surface at –20° and ranging from –68° to 28° with a 3° increment. The magnification was set to give a pixel size around 2 Å and data were collected at a defocus of –2.5 µm. For images collected normally to the lamella surface, the contrast transfer function (CTF) could be fit up to 5.6–7.0 Å (*Figure 4—figure supplement 1*). Tomograms were reconstructed from tilt series taken at synaptic targets by patch tracking methods (*Figure 4C*). Within reconstructed tomograms, we observed the anticipated histological features of neuronal tissue, including myelin-wrapped compartments, microtubules, synaptic vesicles, and mitochondria (*Figure 4D*; *Zuber et al., 2005*). Moreover, there were clusters of AuNPs nearby presynaptic vesicle-containing compartments, and a scattering of more dispersed AuNPs adjacent to the dense clusters. These AuNP features facilitated the identification of synaptic clefts, characterized by a 20–30 nm distance separating the pre- and postsynaptic membranes (*Figure 4C*).

## Anti-GluA2 Fab-AuNP defines AMPARs at glutamatergic synapses

To better understand the spatial distributions of AuNPs bound to a single AMPAR in our tomograms, we first analyzed the positions of AuNPs in a single particle dataset of anti-GluA2 15F1 Fab-AuNP bound to native mouse hippocampus AMPARs. By preparing cryo-EM grids at a low particle density that clearly allowed for the separation of individual receptor complexes, we ensured that nearest neighbor AuNP positions were attributable to AuNPs within, rather than between, receptor complexes. In turn, this allowed us to measure the nearest neighbor projection distance observed between AuNPs bound to an individual AMPAR (*Figure 5A*, *Figure 1—figure supplement 2*). Throughout sample preparation, we included the AMPAR antagonist ZK-200775 (*Turski et al., 1998*) and the positive modulator RR2b (*Kaae et al., 2007*) to stabilize AMPARs in a resting, non-desensitized state (*Sobolevsky et al., 2009*). The distance between the last structured residue (K216) in the heavy chain of each of the 15F1 Fabs in previously published structures of 15F1 Fab-bound hippocampal AMPAR in the resting state is

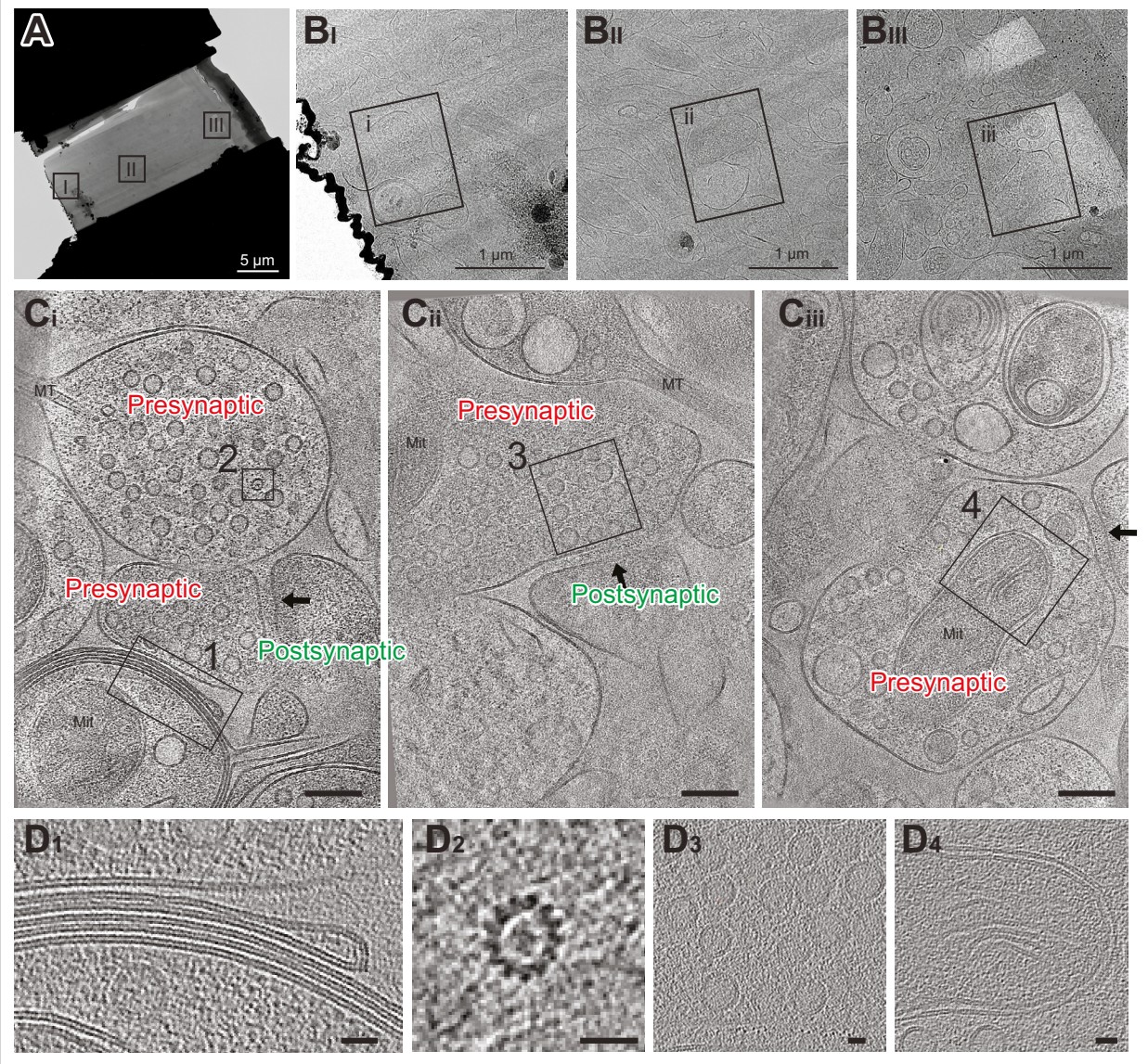

**Figure 4.** Cryo-electron tomography imaging of glutamatergic synapses within brain tissue. (**A**) Medium montage map image of lamella. (**B** I-III) Enlarged images of lamella are indicated in A. (**C** i-iii) Slices from SIRT-filtered tomograms were collected at squares indicated in B. The synaptic cleft in each tomogram is indicated with a black arrow. Mit: mitochondria and MT: microtubule side views. One gold fiducial is visible in **C** iii. Scale bars are 100 nm. Each slice shown is about 10 nm thick. (**D**) Example image of myelin membranes (**D₁**), top-down view of microtubule (**D₂**), synaptic vesicles (**D₃**), and part of a mitochondria (**D₄**). Scale bars are 20 nm. Each slice shown is 10 nm thick.

The online version of this article includes the following figure supplement(s) for figure 4:

**Figure supplement 1.** Power spectra and contrast transfer function (CTF) fitting of tilt series.

approximately 80 Å (*Figure 5—figure supplement 1*). In the 15F1 Fab-AuNP conjugate, the AuNP is covalently bound to a cysteine on the heavy chain located 11 residues after K216, allowing the AuNP as much as 35 Å of displacement from this point. As a result, inter-AuNP distances from Fabs bound to a single receptor can conceivably range from 30 to 150 Å, where an inter-AuNP distance of 30 Å would correspond to two ~30 Å diameter AuNPs immediately adjacent to one another.

The majority of nearest neighbor AuNP projection distances measured in the single particle dataset are in agreement with this range, with major peaks present around 40 and 85 Å (*Figure 5B*). This analysis is based on projection distances, which will generally underestimate the actual distance between AuNPs and will undercount AuNPs overlapping in Z with an inter-AuNP projection distance of less than one AuNP diameter (which are either measured at the minimum picking distance of 30 Å apart

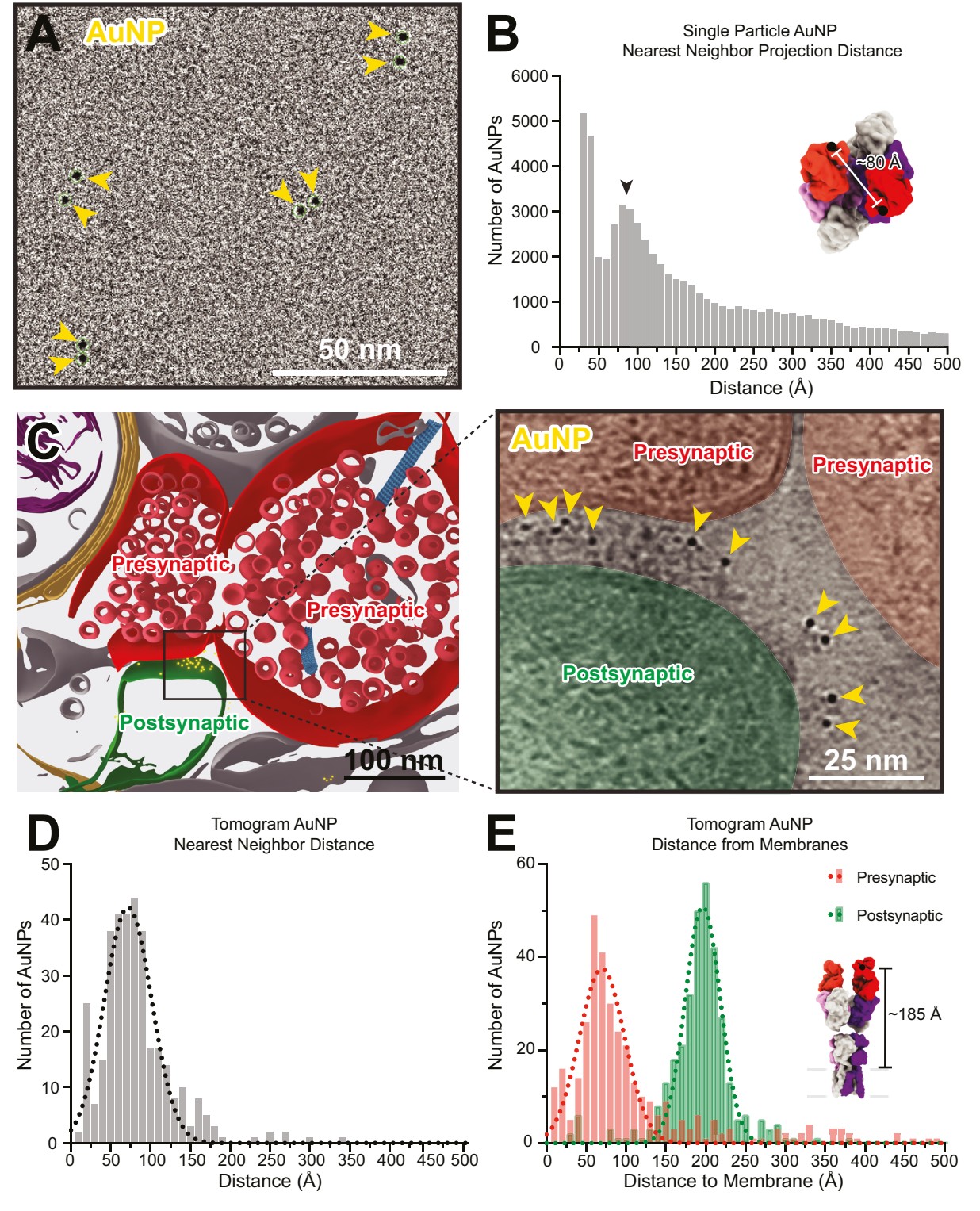

**Figure 5.** Anti-GluA2 Fab-gold nanoparticle (AuNP) labeling defines AMPAR position in single particle cryo-electron and cryo-electron tomography data. (**A**) Snapshot from low defocus single particle cryo-electron microscopy dataset showing four pairs of Fab-AuNPs bound to native mouse hippocampus AMPAR. (**B**) Quantitation of nearest neighbor AuNP projection distances from full ~1000 micrograph dataset from (**A**), overlayed with Gaussian fit of data. Distance between structured C-termini of Fabs in hippocampal AMPAR structure (PDB: 7LDD) shown for reference. (**C**) Segmentation (left) and slice through tomogram (right; denoised using IsoNet) of mouse hippocampus CA1 tissue stained with anti-GluA2 Fab-AuNP, with AuNPs visible in-between the pre-and postsynaptic membranes. Quantitation and Gaussian fits of nearest neighbor AuNP distances (**D**) and

*Figure 5 continued on next page*

*Figure 5 continued*

distance to the closest point on the pre- or postsynaptic membrane (**E**) from five Fab-AuNP labeled synapses. Distance between structured C-terminus of Fab and the putative membrane location in hippocampal AMPAR structure (PDB: 7LDD) shown for reference.

The online version of this article includes the following video and figure supplement(s) for figure 5:

**Figure supplement 1.** Analysis of spatial constraints of 15F1 Fab-gold nanoparticle (AuNP) conjugation.

**Figure 5—video 1.** Representative tomogram and segmentation.

https://elifesciences.org/articles/98458/figures#fig5video1

or detected as a single AuNP). Additionally, this analysis is subject to the effects of preferred orientation bias and aggregation of AMPARs on single-particle cryo-EM grids. Despite these challenges, the histogram of observed inter-AuNP distances has a major peak at 85 Å, which is in overall agreement with the expected 2:1 Fab-AuNP:AMPAR stoichiometry and geometry. These analyses provided confidence that our anti-GluA2 Fab-AuNP conjugate labels hippocampal AMPARs in their native state without overall disruptions to AMPAR structure.

Careful inspection of the synaptic cleft in our tomograms revealed a plethora of strong ~30 Å diameter densities (*Figure 5C*), consistent with the presence of a 15F1-AuNP label. We then used automated tools to create segmentations for the positions of AuNPs and membranes as described in the methods. Nearest neighbor distances of AuNPs at synapses fall mostly within the range of 40–100 Å, fitting to a Gaussian mean of 72±30 Å (*Figure 5D*). AuNPs within tomograms are often densely clustered at putative AMPAR nanodomains (*Nair et al., 2013*), likely contributing to a slight reduction in observed nearest neighbor distances in comparison to the single particle data, in which the low density of AMPARs maximizes analysis of AuNPs bound to individual rather than adjacent AMPARs. The distance between the 15F1 Fab heavy chain K216 residue and the putative membrane in previously published native AMPAR structures is about 185 Å, where the extracellular boundary of the membrane was estimated at a plane defined by four pre-M1 helix residues (*Figure 5—figure supplement 1*). Within 15F1 Fab-AuNP-labeled tomograms, synaptic AuNPs are located at a mean distance of 196±22 Å from the postsynaptic membrane and 70±29 Å from the presynaptic membrane (*Figure 5E*), clearly defining the position of AMPARs at synapses (*Figure 5—video 1*).

## Discussion

The overall goals of the present work are to develop methods to enable the identification and localization of the molecular machinery at chemical synapses in unstained, unfixed native brain tissue slices in order to place mechanisms of synaptic transmission, plasticity, and development on a 3D structural foundation. By harnessing genetically engineered mouse lines, together with cutting of thin brain slices and their subsequent HPF and FIB milling, we have produced substrates for cryo-ET that allow us to target glutamatergic synapses by cryo-fluorescence microscopy at micrometer resolution. Because we have, in parallel, labeled the predominate GluA2 subunit of AMPARs with a highly specific Fab-AuNP conjugate, we can identify glutamatergic synapses and AMPARs at nanometer resolution. This labeling enables clearer identification of synapses and receptor positioning compared to prior studies using cultured neurons or synaptosomes (*Chen et al., 2008*; *Tao et al., 2018*; *Martinez-Sanchez et al., 2021*). These methods will not only facilitate the specific visualization of the molecular machinery of excitatory synapses, but also because they are generalizable, we imagine that similar strategies will be useful for identifying and localizing other key molecules at glutamatergic synapses, such as *N*-methyl-D-aspartate receptors, calcium channels, and vesicle fusion machinery. The approaches outlined here may also be helpful in revealing the macromolecular organization of a wide range of different synapses and cellular structures.

Despite the advances enabled by the approaches described here, there are nevertheless several limitations of the present work. First, while we endeavored to target CA3-CA1 synapses, our ability to accurately and reproducibly identify pyramidal cell-pyramidal cell synapses remains limited, in part because of the resolution limitations of the cryo-fluorescence microscopy and the much smaller field of view in the transmission electron microscopy. Further efforts will be required to optimize correlative targeting and faithful identification of specific synapses, which may include the development and use of additional fluorescence or electron-dense labels. We also are aiming to better understand

the extent to which the labeling of the GluA2-containing receptors is complete and if not, how to 'push' the labeling to completion. The extent to which labeling of GluA2-containing receptors with the Fab AuNP conjugate alters the interactions of the receptors with synaptic proteins, and thus synaptic architecture and function, also warrants further investigation. In addition, the step toward subtomogram averaging of GluA2-containing AMPA receptors, using the AuNP labels, may ultimately be hindered by the flexible linker between the Fab and the AuNP, thus requiring the development of new reagents in which the AuNP is rigidly bound to the Fab, or subtomogram averaging approaches that do not rely upon the AuNP positions. Lastly, a crucial process in the preparation of brain slices for HPF presently involves the inclusion of cryoprotectants in the media, in order to minimize ice formation. At the present time, we do not know how the cryoprotectants alter the functional state of the tissue and the synapses. Further experiments will be required to determine if we can utilize alternative cryoprotectants or reduce their use. If they remain essential, then we aim to better understand their effects on the functional states of the synapses.

Upon resolving the limitations of the methods described here, and working further toward optimization of sample preparation, cryoprotection, targeting, and milling, in combination with the ideal collection of tilt series and image processing, we hope to carry out subtomogram averaging of synaptic receptor complexes. We further aim to identify additional components of the synaptic machinery and pursue their reconstructions, ultimately positioning ourselves to compare the structures of synapses from different brain regions, at selected times during development or in states of disease, thus furthering our understanding of the interrelationships between synapse structure and function.

## Methods

### Dissection and preparation of hippocampal slices

All procedures were performed in accordance with the guidelines of Oregon Health & Science University and the Animal Care and Use Committees approved all of the experimental procedures.

We prepared a mouse line in which we appended mScarlet (*Bindels et al., 2017*), a red fluorescent protein, at the C-terminus of vGlut1 in a C57BL/6 background mouse based on a previously created vGlut1-mVenus mouse line. vGlut1-mScarlet knock-in mouse was generated using CRISPR/Cas-mediated gene targeted strategy in *Slc17a7*. A second mouse line, where the C-terminus of PSD95 is tagged with EGFP in a C57BL/6 background mouse, was a generous gift from Dr. Haining Zhong (*Fortin et al., 2014*). The mice used in the present study are a cross between the vGlut1-mScarlet (HOMO) and PSD95-EGFP (HOMO) lines and were subsequently bred to generate a vGlut1-mScarlet (HET)/PSD95-EGFP (HET) mouse line. Adult male and female mice (8–12 weeks old) were anesthetized with isoflurane and decapitated. Brains were quickly removed and placed in a vibratome (Leica, VT1200). Horizontal brain slices (40 µm) were prepared at room temperature in HEPES base solution containing the following (in mM) 150 NaCl, 2.5 KCl, 2 CaCl$_2$, and 10 HEPES (pH 7.3). AMPA receptor antagonist, ZK-200775 (1 µM), and the positive allosteric modulator, RR2b (1 µM), were included throughout the experiments. From a horizontal brain slice, hippocampal regions were excised and incubated with the GluA2 subunit-specific 15F1 Fab conjugated with AuNP (2 µg/mL or ~40 nM) for 1 hr at room temperature on an orbital shaker (180 rpm), followed by three washes for 15 min each with HEPES base solution at room temperature. Hippocampal slices were further dissected with a scalpel knife to isolate the CA1 region, which was then incubated in a HEPES base solution containing 20% dextran for at least 30 min prior to high-pressure freezing (HPF).

### HPF of hippocampal brain slices

#### Freezing preparation

Planchettes and grids were pretreated such that frozen sample grids could be easily detached from the planchettes after freezing (*Kelley et al., 2022*; *Klykov et al., 2022*). The planchettes (Cu/Au 6 mm, Type B 0.3 mm cavity) were polished on the flat side, using 7000 and 15,000 grit sandpaper, until the surface appeared shiny. The surfaces were then cleaned with a metal polish, Wenol. The flat sides of the planchettes were coated with 1-hexadecene for at least 30 min, which was removed right before use. 200 µL of PEGylated 10 nm gold fiducials (CP11-10-PA-1K-DI water-50, Nanoparts) were spun down at 15,000 rpm for 30 min and the supernatant was removed. After adding 200 µL of HEPES

base solution, the centrifugation step was repeated and the supernatant was again removed. Next, 50 µL of HEPES base solution containing cryoprotectants (20% dextran and 5% sucrose) was added. This mixture was stored at room temperature until use. Right before freezing samples, 200 mesh gold extra thick carbon grids (CF200-Au-ET; EMS) were glow discharged with a PELCO easiGlow unit (Ted Pella) for 30 s at 15 mA current with the grid bar sides up.

## HPF of brain slice sample

The sample and 6 mm flat specimen planchettes were prepared for HPF at the loading station of a Leica EM ICE. First, a bottom planchette was placed on a lower half cylinder. Two CA1 tissue slices were placed on the grid bar side of a grid with apical dendrites facing toward the center of the grid, and a sample grid was placed on the bottom planchette sample side up. Right before HPF, a small volume (1.5–3 µL) of 10 nm diameter PEGylated gold fiducials (Nanoparts) in HEPES base solution supplemented with 20% dextran and 5% sucrose was applied on top of the sample grid. A second planchette was then placed on top of the sample, flat side down, making sure to not introduce an air bubble. The assembled cartridge was then subjected to the automated HPF cycle that involves pressurization to approximately 2050 bar (210 kPa) and cooling to –196 °C within 10 ms. After HPF, samples were transferred into the auto grid clipping station to disassemble the grid from the planchettes. Sample grids were then clipped using a cryo-FIB autogrid ring and C-clip, with the carbon layer facing the autogrid ring and the sample side facing the C-clip.

## Cryogenic focused ion beam (Cryo-FIB) milling

The HPF hippocampal brain slice grids were milled on an Aquilos 2 cryo-FIB/SEM microscope (Thermo Fisher Scientific). Scanning electron microscope (SEM) images of the whole grid were taken using a nominal magnification of 350 x and an acceleration voltage of 2 kV with a beam current of 13 pA and a dwell time of 1 µs. Sputter coating with a platinum layer (30 mA current for 15 s at 10 Pa pressure) and a gas injection system (GIS) deposition of a trimethyl(methylcyclopentadienyl) platinum layer (2 min) were applied before FIB milling. The brain slices were placed on the grid such that the Schaffer collaterals, where CA3 axons synapse onto CA1 apical dendrites, were located near the center of the grids. We picked lamella positions in the middle of the grids to increase the likelihood of imaging these CA3 – CA1 synapses. The HPF brain tissue lamellae were milled using the 'waffle' method (*Kelley et al., 2022*; *Klykov et al., 2022*). Briefly, two rectangular trench cuts were milled at 15 nA ion beam current to clear the front and back of the target lamella site. After making trench cuts at each site, a preemptive 'pre clean' step was utilized to further remove the bottom portion of the sample. The precleans were completed by using incremental angles; starting at 40°, then 30°, followed by 20°. Following precleans, the notch patterns were milled at the side of the final lamella sites. Automated milling was performed at the milling angle of 20° by progressively reducing the ion beam current from 1 nA to 30 pA to produce a final lamella thickness of approximately 150–250 nm. After automated milling, the grids were tilted and manually polished at both +0.5° (20.5°) and –0.2°(19.8°) with an ion beam current of 50 or 30 pA. The milled grids were retrieved from the Aquilos 2 and stored in liquid nitrogen until they were imaged.

## Cryo-confocal imaging of milled sample grids

The milled grids were loaded onto a confocal microscope (Zeiss LSM980) fitted with a CMS196 cryo-stage (Linkam). The chamber and the grid stages were maintained below –185°C throughout imaging. First, the whole grid was imaged via widefield fluorescent light using 5 x and 10 x objectives. Fluorescence excitation was elicited using an X-Cite Xylis LED light (Excelitas technologies) and emission was detected using an Axiocam 506 camera (Zeiss). For higher resolution, confocal laser scanning microscope images were taken using a 100 x (0.75 NA) objective with and without an Airyscan module. Imaging settings, such as pixel resolution and pinhole diameter, were set according to the Zen 3.7 software for optimal Nyquist-based resolution. Laser power settings and dwell times were set slightly higher and longer than those used for room temperature experiments, as the photobleaching is reduced at cryogenic temperature (*Kaufmann et al., 2014*). At this sample thickness, the laser exposure did not cause significant heating and thus the sample was not devitrified. At each lamella, a cropped imaging area was designated to include only the lamella area and a Z-stack range was set from the milling edge to the end of the lamella with a 1 µm interval Z-spacing. Approximately

eight Z-stack images were captured and compiled into a maximum intensity projection to visualize an entire single lamella at –20° tilt.

## TEM tilt series acquisition

Tilt series were acquired using a 300 keV FEI Titan Krios cryo transmission electron microscope, equipped with a spherical aberration corrector, a Gatan BioContinuum energy filter (6 eV energy slit width), and a K3 direct detector (Gatan, CA). Medium montage map images from each lamella were collected with a nominal magnification of 2,250 x with a calibrated pixel size of 60.3 Å. Tilt series were collected using a dose-symmetric scheme (*Hagen et al., 2017*) starting at –20° and ranging from –68° to 28°, with a 3° increment using SerialEM (*Mastronarde, 2005*). Data were acquired with a magnification of 33,000 (calibrated pixel size of 2.06 Å) on a K3 detector with a total electron dose of 150 e⁻/Å² and a target defocus of –2.5 μm. At each tilt, a movie of eight frames was collected.

## Tomogram reconstruction with patch tracking

Movies were motion corrected using MotionCor2 (*Zheng et al., 2017*) without dose weighting and binned to a pixel size of 2.5 Å. Initially, tomograms were reconstructed using the fiducial-free alignment software AreTomo (*Zheng et al., 2022*). The initial tomograms were inspected individually to determine which tomograms warranted further image processing. Tilt series with low-quality ice, large ice contamination, or with no discernable Fab AuNP particles, were discarded. Several good-quality tilt series were chosen to go through tomogram reconstruction in Etomo using patch tracking (*Mastronarde and Held, 2017*). Tilt series were binned by 4 for patch tracking alignment and CTF corrected with Ctfplotter within Etomo (*Mastronarde, 2024*). The final tomograms were binned to a final pixel size of 10 Å.

## AuNP synthesis

Thiolate-protected AuNPs were prepared as previously described (*Azubel and Kornberg, 2016*; *Elferich et al., 2021*) using a 7:1 molar ratio of 3-mercaptobenzoic acid (3-MBA):HAuCl₄ to produce uniform nanoparticles with diameters of approximately 3 nm. Briefly, 1 mL of 84 mM 3-MBA was mixed with 0.4 mL of 28 mM HAuCl₄ in a 15 mL plastic (Falcon) tube, and 3.5 mL of water was added, producing a white precipitate. NaOH was added dropwise to 100 mM (49.5 μL of 10 M NaOH), resolubilizing the precipitate. The reaction was mixed by rotation at room temperature for at least 16 hr, after which the solution was moved to a 50 mL plastic tube and 28.7 mL of 27% methanol was added. NaBH₄ was added to a final concentration of 2 mM, using approximately 450 μL of a 150 mM stock solution of NaBH₄, and the reaction was continued by rotation at room temperature for 4.5 hr. AuNPs were precipitated by adding NaCl to 100 mM (694 μL of 5 M NaCl added to a 34 mL reaction), followed by the addition of 80 mL of methanol. The reaction solution was distributed into three 50 mL plastic tubes and centrifuged at 4100 *g* for 20 min. Pellets were gently washed with 75% aqueous methanol, combined into a single plastic tube, and centrifuged at 4100 *g* for 20 min. The remaining methanol was removed and the pellet was dried overnight in a desiccator. The dried pellet was resuspended in water to ~10 mg/mL, resulting in a black homogeneous solution. An aliquot of AuNPs was diluted 1:1 with 10% aqueous glycerol in 0.2 X TBE (20 mM Tris base, 20 mM boric acid, 0.5 mM EDTA) and run on a 10% glycerol, 12% PAGE gel in 0.2 X TBE running buffer to assess AuNP size and homogeneity.

## Anti-GluA2 15F1 Fab-AuNP conjugation

An anti-GluA2 15F1 Fab (*Giannone et al., 2010*) construct was designed with an extended heavy chain fragment containing a single hinge cysteine for AuNP conjugation (C-terminal sequence: KVDK-KIVPRDAGAKPC) followed by a C-terminal Twin Strep-tag. This construct, deemed 15F1FabxC-2xStr, was expressed in Sf9 insect cells by baculovirus transduction (*Yu et al., 2021*) and purified over Strep-Tactin Superflow resin in T150 buffer (20 mM Tris-Cl pH 8.0, 150 mM NaCl). Approximately 400 μg of 15F1FabxC-2xStr was diluted to 20 μM in TBE (100 mM Tris base, 100 mM boric acid, 2.5 mM EDTA) with 2 mM tris(2-carboxyethyl)phosphine (TCEP) and incubated at 37 °C for 60 min. The reduced Fab was purified by size exclusion chromatography and concentrated at 1 mg/mL. To determine optimal AuNP conjugation conditions, a range of Fab:AuNP ratios were tested in small-scale conjugation reactions. Fab (1 mg/mL; 1–8 μL) and AuNP (10 mg/mL; 1–8 μL) were mixed in TBE at a total volume

of 10 μL (Fab:AuNP volumes of 8:1, 4:1, 2:1, 1:1, 1:2, 1:4, or 1:8 μL) and incubated at 37 °C for 30 min before diluting 1:1 with 10% glycerol and loading 15 μL onto a 10% glycerol, 12% PAGE gel. A 2:1 Fab:AuNP ratio was determined to be the optimal condition for conjugation, and the reaction was scaled up by incubating 300 μL of 1 mg/mL 15F1FabxC-2xStr with 150 μL of 10 mg/mL AuNP and 50 μL 10 X TBE at 37 °C for 30 min. After adding 25 μL of glycerol, the entire reaction was run on a 10% glycerol, 12% PAGE gel (*Figure 1—figure supplement 1C*; *Azubel and Kornberg, 2016*). Bands corresponding to the 1:1 Fab:AuNP species were cut out, chopped into small pieces, and incubated overnight at 4 °C in TBE buffer to extract the 15F1 Fab-AuNP conjugate. The supernatant containing the 15F1 Fab-AuNP conjugate was concentrated to 400 μL and incubated with 0.5 mM of PEG550-SH (3.3 μL of 60 mM PEG550-SH) for 60 min at 37 °C. An optimal PEG550-SH concentration of 0.5 mM was determined to PEGylate the AuNPs without displacing the Fab (*Figure 1—figure supplement 1D*). Excess PEG was removed by serial concentration and the final PEGylated Fab-AuNP was stored at 4 °C. The final concentration of 15F1 Fab-AuNP was estimated by comparison to Fab standards on a denaturing gel.

## Hippocampal AMPAR purification with 15F1 Fab-AuNP

Hippocampi from 10 male and four female adult vGlut1-mScarlet mice (total tissue mass = 675 mg) were resuspended in chilled homogenization buffer (20 mM Tris-HCl pH 8.0, 150 mM NaCl, 0.8 μM aprotinin, 2 μg/mL leupeptin, 2 μM pepstatin A, 2 μM ZK-200775, 2 μM JNJ-55511118, 50 μM RR2b) at 7.5 mL buffer per gram of tissue. Tissue was homogenized with 10 strokes on a Dounce homogenizer, then diluted 1:1 with solubilization buffer (homogenization buffer plus 4% w/v digitonin) and incubated at 4 °C for 15 min on a nutator. The solubilized tissue was centrifuged at 4000 *g* for 3 min at 4 °C and filtered through a 0.22 μm filter. Anti-GluA2 15F1 Fab-AuNP was added to the sample to a final concentration of approximately 40 nM and the sample was 'nutated' at 4 °C for 15 min. The solution was then passed over 5 mL of StrepTactin Superflow resin equilibrated in purification buffer (20 mM Tris-HCl pH 8.0, 150 mM NaCl, 2 μM ZK-200775, 2 μM JNJ-55511118, 1 μM RR2b, 0.075% w/v digitonin) by gravity flow and eluted with purification buffer +5 mM desthiobiotin. The resulting native AMPAR bound to 15F1 Fab-AuNP was further purified by size exclusion chromatography, using an HPLC. The absorbance at 280 nm was monitored and peak fractions corresponding to the 15F1 Fab-AuNP bound native AMPAR species were collected by hand and concentrated to 30 μL. The concentrated sample was diluted 1:1 in purification buffer and 3 μL were applied to glow-discharged Quantifoil R2/1 Cu 200 mesh grids covered with a 2 nm continuous carbon film. The grids were blotted and plunge frozen by submersion into a 35/65% ethane/propane mix using an FEI Vitrobot set to 4 °C, 100% humidity, 30 s wait time, 2.5 s blot time, and 0 blot force.

## Single particle cryo-EM data collection and processing

Single particle cryo-EM grids with native mouse hippocampal AMPAR bound to 15F1 Fab-AuNP were imaged on a 200 kV Thermo Fisher Glacios microscope equipped with a Gatan K3 Summit direct electron detector at a magnification of 45,000 x (pixel size of 0.45 Å in super-resolution mode) and a total dose of 50 e/Å$^2$. Data were collected using the SerialEM multi-shot 3×3 pattern with each movie containing 50 frames collected over a total exposure time of 2.9 s. For analysis of nearest neighbor inter-AuNP distances, a 'low defocus' dataset of approximately 1000 movies was collected at a defocus of –1.0 to –1.2 μm, a range in which the automated AuNP bead picking software 'imod-findbeads' (*Mastronarde and Held, 2017*) was most accurate at picking AuNPs and avoiding background signal, as judged by manual analysis of picking results. For optimal visualization of receptor density along with AuNPs, several movies were collected at a higher defocus range of –4.5 to –5.0 μm. All movies were motioncorrected in cryoSPARC with patch motion correction (*Punjani et al., 2017*).

## Tomogram segmentation and AuNP distance analyses

A custom Python script was written to identify the 2D coordinate positions of AuNPs from all motion-corrected single particle cryo-EM movies in the 'low defocus' dataset and calculate the nearest neighbor distance to the next closest AuNP for each AuNP position. The 'imodfindbeads' command within IMOD (version 4.12.56) was used for the automated identification of AuNPs within the micrographs (*Mastronarde and Held, 2017*).

To analyze inter-AuNP and AuNP-membrane distances within tomograms, membranes in each tomogram were segmented with membrain-seg (*Lamm et al., 2022*). The 'findbeads3D' command within IMOD (version 4.12.56) was used to identify the 3D coordinate positions of all AuNPs from within a manually specified subregion of the tomogram, using the following options: BeadSize = 1.5, ThresholdForAveraging = 20, and StorageThreshold = 0.5. An optimal thresholding value to accurately pick AuNPs over the background was manually determined for each analyzed subregion. The identity of presynaptic, postsynaptic, and myelin membranes were manually annotated and microtubules were traced by hand. The distances to the nearest AuNP neighbor and point on the pre- or postsynaptic membrane were calculated for each identified AuNP and fit to Gaussian distributions by nonlinear regression analysis using GraphPad Prism 10.1.1. Segmentations were rendered using Blender and the MolecularNodes plugin (*Johnston et al., 2022*). Where indicated, tomograms were denoised using CryoCare (*Buchholz et al., 2019*) or IsoNet (*Liu et al., 2022*).

## Acknowledgements

We thank Dr. Lauren Ann Metskas (LAM; Purdue) for advice, education, and training related to cryo-electron tomography methods, Dr. Chang Sun for assistance with software installation and computer workstations, Dr. Nelson Spruston (HHMI Janelia) and the HHMI Janelia transgenic facility for their support in generating the precursor line of the PSD95-CreNABLED2 mouse, Rachel Courtney for assistance with manuscript preparation, Mark Mayer and LAM for comments on the manuscript, and Dr. John T Williams for sharing a vibratome. We acknowledge the generous support and use of the HHMI Janelia Cryo-EM facility for FIB milling on an Aquilos2 and data collection on Krios microscopes, which were operated by Drs. Shixin Yang, Rui Yan, and James Jung (HHMI Janelia), and the OHSU MMC facility for data collection on the Glacios with assistance from Erin Stempinski. This research was support by the National Institute of Neurological Disorders and Stroke (5R01NS038631) to EG. CJS is supported by grant number CA253730 from the National Cancer Institute at the National Institutes of Health. EG is an investigator of the Howard Hughes Medical Institute and gratefully acknowledges the generous support of Jennifer and Bernard LaCroute.

## Additional information

### Funding

| Funder | Grant reference number | Author |
| --- | --- | --- |
| National Cancer Institute | CA253730 | Cathy Spangler |
| Howard Hughes Medical Institute | | Eric Gouaux |
| Jennifer and Bernard LaCroute | | Eric Gouaux |
| National Institute of Neurological Disorders & Stroke | NS038631 | Eric Gouaux |

The funders had no role in study design, data collection and interpretation, or the decision to submit the work for publication.

### Author contributions

Aya Matsui, Prepared samples and performed the experiments, processed and visualized data, designed the project and wrote the manuscript; Cathy Spangler, Prepared samples and performed the experiments, processed and visualized data, designed the project and wrote the manuscript; Johannes Elferich, Processed and visualized data; Momoko Shiozaki, Performed cryo-FIB milling of samples; Nikki Jean, Performed cryo-FIB milling of samples; Xiaowei Zhao, Assisted with tilt series data acquisition; Maozhen Qin, Provided the PSD95-EGFP mouse line; Haining Zhong, Provided the PSD95-EGFP mouse line; Zhiheng Yu, Assisted with tilt series data acquisition; Eric Gouaux, Designed the project and wrote the manuscript

## Author ORCIDs

Aya Matsui  https://orcid.org/0000-0003-4437-8278
Cathy Spangler  https://orcid.org/0000-0003-2334-3957
Johannes Elferich  https://orcid.org/0000-0002-9911-706X
Haining Zhong  https://orcid.org/0000-0002-7109-4724
Eric Gouaux  https://orcid.org/0000-0002-8549-2360

## Ethics

This study was performed in strict accordance with the recommendations in the Guide for the Care and Use of Laboratory Animals of the National Institutes of Health. All of the animals were handled according to approved Institutional Animal Care and Use Committee (IACUC) protocols of Oregon Health and Science University. The protocol was approved by the IACUC of Oregon Health and Science University (IP00000905). All euthanasia was performed under isoflurane anesthesia, and every effort was made to minimize suffering.

Reviewer #1 (Public review): https://doi.org/10.7554/eLife.98458.3.sa1
Reviewer #2 (Public review): https://doi.org/10.7554/eLife.98458.3.sa2
Author response https://doi.org/10.7554/eLife.98458.3.sa3

---

# Additional files

## Supplementary files

• MDAR checklist

## Data availability

Tomograms have been deposited to EMDB under accession codes EMD-44174, EMD-44175, and EMD-44176. The corresponding raw data has been deposited to EMPIAR under accession code EMPIAR-11984.

The following datasets were generated:

| Author(s) | Year | Dataset title | Dataset URL | Database and Identifier |
|---|---|---|---|---|
| Matsui A, Spangler CJ, Elferich J, Gouaux E | 2024 | Cryo-electron tomographic investigation of native hippocampal glutamatergic synapses - Tomogram 1 | https://www.ebi.ac.uk/emdb/EMD-44174 | Electron Microscopy Data Bank, EMD-44174 |
| Matsui A, Spangler CJ, Elferich J, Gouaux E | 2024 | Cryo-electron tomographic investigation of native hippocampal glutamatergic synapses - Tomogram 2 | https://www.ebi.ac.uk/emdb/EMD-44175 | Electron Microscopy Data Bank, EMD-44175 |
| Matsui A, Spangler CJ, Elferich J, Gouaux E | 2024 | Cryo-electron tomographic investigation of native hippocampal glutamatergic synapses - Tomogram 3 | https://www.ebi.ac.uk/emdb/EMD-44176 | Electron Microscopy Data Bank, EMD-44176 |
| Matsui A, Spangler CJ, Elferich J, Gouaux E | 2024 | Cryo-electron tomographic investigation of native hippocampal glutamatergic synapses | https://www.ebi.ac.uk/empiar/EMPIAR-11984/ | Electron Microscopy Public Image Archive, EMPIAR-11984 |

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
