## [Editor Report · eLife assessment]

This **fundamental** study demonstrates a novel method for imaging glutamate receptors in situ via cryo-ET. The use of cutting-edge methods is well-described and is **compelling**. This paper is broadly relevant to biophysicists and neuroscientists.

---

## [Referee Report · Reviewer #1 (Public review)]

Summary:

Matsui et al. present an experimental pipeline for visualizing molecular machinery of synapses in the brain, which includes numerous techniques, starting with generating labeled antibodies and recombinant mice, continuing with HPF and FIB milling and finishing with tilt series collection and 3D image processing. This pipeline represents a breakthrough in preparation of brain tissue for high resolution imaging and can be used in future tomographic research to reconstruct molecular details of synaptic complexes as well as pre- and post-synaptic assemblies. This methodology can also be adapted for a broader range of tissue preparations and signifies the next step towards better structural understanding of how molecular machineries operate in natural conditions.

Strengths:

The manuscript is very well written, contains a detailed description of methodology, provides nice illustrations and will be an outstanding guide for future research.

Weaknesses:

None noted.

---

## [Referee Report · Reviewer #2 (Public review)]

Summary

The authors present a method that allows for the identification and localization of molecular machinery at chemical synapses in unstained, unfixed native brain tissue slices. They believe that this approach will provide a 3D structural basis for understanding different mechanisms of synaptic transmission, plasticity, and development. To achieve this, the group used genetically engineered mouse lines and generated thin brain slices that underwent high-pressure freezing (HPF) and focused ion beam (FIB) milling. Utilizing cryo-electron tomography (cryo-ET) and integrating it with cryo-fluorescence microscopy, they achieved micrometer resolution in identifying the glutamatergic synapses along with nanometer resolution to locate AMPA receptors GluA2-subunits using Fab-AuNP conjugates. The findings are summarized with detailed examples of successfully prepared substrates for cryo-ET, specific morphological identification and localization, and the detailed structural organization of excitatory synapses, including synaptic vesicle clusters close to the postsynaptic density and in the cleft.

Strengths

The study advances previous work that used cultured neurons or synaptosomes. Combining cryo-electron tomography (cryo-ET) with fluorescence-guided targeting and labeling with Fab-AuNP conjugates enabled the study of synapses and molecular structures in their native environment without chemical fixation or staining. This preserves their near-native state, offering high specificity and resolution. The methods developed are mostly generalizable, allowing adaptation for identifying and localizing other key molecules at glutamatergic synapses and potentially useful for studying a variety of synapses and cellular structures beyond the scope of this research.

Weaknesses

The preparation and imaging techniques are complex and require highly specialized equipment and expertise, potentially limiting their accessibility and widespread adoption.

Additionally, the methods might need further modifications/tweaks to study other types of synapses or molecular structures effectively.

The reliance on genetically engineered mouse lines and monoclonal, high-affinity antibodies/Fab fragments to specifically label receptors/proteins would limit the wider employment of these methods.

---

## [Author Response]

The following is the authors’ response to the original reviews.

**Public Reviews:**

**Reviewer #1 (Public Review):**
Summary:Matsui et al. present an experimental pipeline for visualizing the molecular machinery of synapses in the brain, which includes numerous techniques, starting with generating labeled antibodies and recombinant mice, continuing with HPF and FIB milling, and finishing with tilt series collection and 3D image processing. This pipeline represents a breakthrough in the preparation of brain tissue for high-resolution imaging and can be used in future tomographic research to reconstruct molecular details of synaptic complexes as well as pre- and post-synaptic assemblies. This methodology can also be adapted for a broader range of tissue preparations and signifies the next step towards a better structural understanding of how molecular machineries operate in natural conditions.Strengths:The manuscript is very well written, contains a detailed description of methodology, provides nice illustrations, and will be an outstanding guide for future research.Weaknesses:None noted.
**Reviewer #2 (Public Review):**
Summary:The authors present a method that allows for the identification and localization of molecular machinery at chemical synapses in unstained, unfixed native brain tissue slices. They believe that this approach will provide a 3D structural basis for understanding different mechanisms of synaptic transmission, plasticity, and development. To achieve this, the group used genetically engineered mouse lines and generated thin brain slices that underwent high-pressure freezing (HPF) and focused ion beam (FIB) milling. Utilizing cryo-electron tomography (cryo-ET) and integrating it with cryo-fluorescence microscopy, they achieved micrometer resolution in identifying the glutamatergic synapses along with nanometer resolution to locate AMPA receptors GluA2-subunits using Fab-AuNP conjugates. The findings are summarized with detailed examples of successfully prepared substrates for cryo-ET, specific morphological identification and localization, and the detailed structural organization of excitatory synapses, including synaptic vesicle clusters close to the postsynaptic density and in the cleft.Strengths:The study advances previous work that used cultured neurons or synaptosomes. Combining cryo-electron tomography (cryo-ET) with fluorescence-guided targeting and labeling with Fab-AuNP conjugates enabled the study of synapses and molecular structures in their native environment without chemical fixation or staining. This preserves their near-native state, offering high specificity and resolution. The methods developed are generalizable, allowing adaptation for identifying and localizing other key molecules at glutamatergic synapses and potentially useful for studying a variety of synapses and cellular structures beyond the scope of this research.WeaknessesThe preparation and imaging techniques are complex and require highly specialized equipment and expertise, potentially limiting their accessibility and widespread adoption.Additionally, the methods might need further modifications/tweaks to study other types of synapses or molecular structures effectively.The reliance on genetically engineered mouse lines may again impact the generalizability of the findings.Similarly, the requirement of monoclonal, high-affinity antibodies/Fab fragments to specifically label receptors/proteins would limit the wider employment of these methods.
**Recommendations for the authors:**

**Reviewer #1 (Recommendations For The Authors):**
Matsui et al. present an experimental pipeline for visualizing the molecular machinery of synapsis in the brain, which includes numerous techniques, starting with generating labeled antibodies and recombinant mice, continuing with HPF and FIB milling, and finishing with tilt series collection and 3D image processing. This pipeline represents a breakthrough in the preparation of brain tissue for high-resolution imaging and can be used in future tomographic research to reconstruct molecular details of synaptic complexes as well as pre- and post-synaptic assemblies. This methodology can also be adapted for a broader range of tissue preparations and signifies the next step towards a better structural understanding of how molecular machineries operate in natural conditions.The manuscript is very well written, contains a detailed description of methodology, provides nice illustrations, and will be an outstanding guide for future research. I only have a few suggestions to further improve this excellent manuscript.The labeling experiment in Supplementary Figure 3 may have a limitation in the accessibility of certain "narrow" regions to 15F1Fabs (both JF646 and AuNP labeled). Would that be more correct to refer to the labeling of accessible GluA2-containing AMPARs rather than the majority of these receptors in the tissue (lines 180-183)?

The text has been modified to reference “accessible GluA2-containing AMPARs”

Minor comments:(1) Lines 38-39. "natively derived" appears to be unnecessary here and can be deleted.

Done

(2) Line 153. Please specify the 20% dextran cryoprotectant.

Done.

(3) Lines 155-157. Please label the stratum radiatum and stratum lacunosum-moleculare in Figure 3B.

Done

(4) Figures 1C, 2B, 5B, 5D-E. Missing units for Y-axes.

Done

(5) Supplemental Figure 1. Please add band annotation.

Done

(6) Supplemental Figure 3. Scale bars are missing.

Done

(7) Supplemental Video 1 does not play.

The video file has been corrected.

**Reviewer #2 (Recommendations For The Authors):**
My congratulations to the authors for undertaking this challenging work.Major concerns that need to be addressed:It's unclear if the anti-GluA2 15F1 Fab-AuNP conjugate would affect the receptor clustering and localization on the synaptic membranes. It binds at the distal end, which is likely to impact its interactions with other synaptic proteins, which may affect the synaptic organization and function.

Concern addressed in the ‘Discussion’ section.

The hippocampal slices were treated with the anti-GluA2 15F1 Fab-148 AuNP conjugate for 1 hour at room temperature. It might be helpful to discuss the potential affects of Fab-AuNp on synaptic function. It has been demonstrated previously that introducing binders of the receptors ectodomains can affect synaptic function.

Concern also addressed in the ‘Discussion’ section.

Kunimichi Suzuki et al. Science369,eabb4853(2020).DOI:10.1126/science.abb4853 https://patents.google.com/patent/US20230192810A1/en